# CONTRASTIVE POST-TRAINING LARGE LANGUAGE MODELS ON DATA CURRICULUM

## ABSTRACT

Alignment serves as an important step to steer large language models (LLMs) towards human preferences. In this paper, we explore contrastive post-training techniques for alignment by automatically constructing preference pairs from multiple models of varying strengths (e.g., InstructGPT, ChatGPT and GPT-4). We carefully compare the contrastive techniques of SLiC and DPO to SFT baselines and find that DPO provides a step-function improvement even after continueing SFT saturates. We also explore a data curriculum learning scheme for contrastive post-training, which starts by learning from "easier" pairs and transitioning to "harder" ones, which further improves alignment. Finally, we scale up our experiments to train with more data and larger models like Orca. Remarkably, contrastive post-training further improves the performance of Orca, already a state-of-the-art instruction learning model tuned with GPT-4 outputs, to exceed that of ChatGPT.[1]

## 1 INTRODUCTION

The rapid evolution of Large Language Models (LLMs) has ushered in a new era of natural language processing capabilities. These models, when scaled to billions of parameters and pretrained over trillions of text tokens, demonstrate unprecedented proficiency in a wide array of tasks (Brown et al., 2020; Chowdhery et al., 2022). Various *post-training* procedures like supervised instruction tuning and Reinforcement Learning from Human Feedback (RLHF) fine-tune pretrained LLMs to better align with human expectations and preferences (Ouyang et al., 2022; OpenAI, 2023; Touvron et al., 2023a). This additional alignment procedure is crucial, because the pretraining objective of essentially predicting the next token in a text sequence is known to produce LLMs whose outputs are at times incorrect, irrelevant, or unsafe (Bai et al., 2022a).

Traditionally, these post-training techniques rely on human preference annotations to inform an LLM which behaviors it ought to adopt in the scenario at hand. For instance, RLHF fits a reward model on these preference pairs, against which a LLM policy is then optimized (Ziegler et al., 2019; Bai et al., 2022a; Touvron et al., 2023b). However, such human feedback is expensive to obtain and often noisy (Stiennon et al., 2020; Ouyang et al., 2022; Bai et al., 2022a).

To align an LLM without human feedback, other methods such as Reinforcement Learning from AI Feedback (RLAIF) harvest preference signals via automatic feedback from another LLM (Lee et al., 2023; Bai et al., 2022b). However, studies have found AI feedback has a low agreement rate with humans (Perez et al., 2022; Casper et al., 2023b; Lee et al., 2021). Also, these methods suffer from the same drawbacks as RLHF, such as reward hacking (Skalse et al., 2022).

Recently, certain *contrastive post-training* techniques such as Sequence Likelihood Calibration (SLiC) and Direct Preference Optimization (DPO) offer enticing alternatives to RLHF (Zhao et al., 2023b;a). For instance, DPO is proven to optimize the same objective as RLHF. But instead of optimizing against a reward model, it works by increasing the LLM's relative probability of generating the preferred output over the unfavorable one — making it much simpler to implement (Rafailov et al., 2023). The difference between the post-training methods is illustrated in Figure 1.

In this work, we study what we believe is a strong connection between *contrastive post-training* and RLAIF: one can employ LLMs to automatically generate preference pairs which can then

---

[1]We will make the code and checkpoints publicly available upon acceptance.

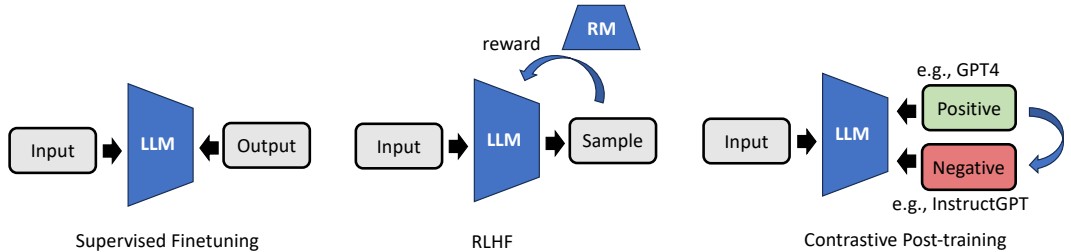

Figure 1: Difference betwen SFT, RLHF, and contrastive post-training. For SFT, the model optimizes the negative log-likelihood for the next token. RLHF samples an output from the LLM and use a reward model to provide feedback for PPO to update the LLM. For contrastive post-training, a contrastive loss is used to steer the model towards preferred outputs.

be optimized directly via contrastive objectives like DPO. However, without feedback from human annotations, LLM-feedback, or a reward model to distinguish them, the key question becomes how to automatically construct pairs that 1) contain meaningful directional signal on a per-example basis and 2) in aggregate adhere to the values and principles that humans expect.

This paper explores a simple yet effective answer to this question: contrast outputs from LLMs of varying sizes and capabilities, as motivated in Table 1. We automatically construct training pairs of responses generated from InstructGPT (Ouyang et al., 2022), ChatGPT, and GPT-4 (OpenAI, 2023) as demonstrations of desirable and undesirable behaviors. We believe this choice provides a solid foundation to better understand the efficacy of various contrastive training techniques when it comes to "bridging the gap" between stronger and weaker models. On a more general level, we wish to apply our findings to improve model distillation (Hinton et al., 2015), i.e., preserve the quality of larger, more capable models in a smaller target model which is cheaper and faster to deploy at scale, as explored in many recent works (Chiang et al., 2023; Xu et al., 2023b; Geng et al., 2023).

Table 1: The win rates of GPT models against each other on the official Alpaca Eval leaderboard motivate our automatic pair construction.

| Model | vs. | Win Rate |
|---------|-------------|----------|
| GPT-4 | InstructGPT | 95.3% |
| GPT-4 | ChatGPT | 83.5% |
| ChatGPT | InstructGPT | 89.4% |

We show through carefully crafted experiments that contrastive post-training techniques maintain a step-function advantage over continuous supervised fine-tuning, which holds even at larger scales of models and training examples. For example, a key result of our study is that enhancing Orca (Mukherjee et al., 2023) — already a state-of-the-art instruction learning model — with DPO over pairs of GPT4-vs-InstructGPT is more beneficial than additional supervised fine-tuning on only the GPT-4 outputs, all else being equal. In fact, the contrastive fine-tuning of Orca is preferred 55%-45% against ChatGPT in head-to-head comparison on the Alpaca Eval benchmark.

Additionally, we structure how and when the model is exposed to various types of pairs in the style of curriculum learning (Bengio et al., 2009; Soviany et al., 2022). We discover that reordering the training data to start from "easy pairs" and warm up to "harder pairs" leads to considerable performance improvements.

To summarize, our contributions are as follows:

1. We propose a new automatic setting for contrastive post-training that improves performance of LLMs without human-, AI-, or reward model-feedback.

2. We explore several curriculums for SFT and DPO. We discover that performance of DPO can be further improved by simply reordering the data.

3. We verify the effectiveness of our approach holds on scaled-up experiments on a state-of-the-art instruction-following model Orca.

## 2 RELATED WORKS

Improving downstream performance of Large Language Models (LLMs) and aligning them with user preference and designed intents are important to deployment and applications. This can be achieved by fine-tuning these models on responses written by humans or generated with human-written labels and templates. Previous works have applied supervised fine-tuning (SFT) on both instruction data (Sanh et al., 2022; Wei et al., 2022; Chung et al., 2022; Taori et al., 2023; Peng et al., 2023) and dialogue data (Chiang et al., 2023; Xu et al., 2023b; Geng et al., 2023). Although SFT can successfully adapt an LLM to instruction learning or chatting, the model can be further improved by post-training (Ouyang et al., 2022) to meet human preference. A straightforward solution to optimize the human preference is to use reinforcement learning. Reinforcement Learning with Human Feedback (RLHF, Ziegler et al., 2019) first trains a Bradley-Terry reward model (Bradley & Terry, 1952) on human-labeled preference pairs. Then, it samples output from the model and scores the output with the reward model. A reinforcement learning algorithm, such as Proximal Policy Optimization (PPO, Schulman et al., 2017) is used to optimize the language model for better rewards. RLHF has seen successful applications in downstream tasks (Kreutzer et al., 2018; Stiennon et al., 2020). However, RLHF methods are infamous for their instability, inefficiency, reward misgeneralization and hacking (Casper et al., 2023a; Skalse et al., 2022).

Recently, there are studies proposing methods for post-training without reinforcement learning. These methods optimize human preference with human-labeled contrastive pairs. FeedMe (OpenAI, 2022) samples model output multiple times and fine-tunes on the best response picked by human labelers. Sequence Likelihood Calibration (SLiC, Zhao et al., 2023b;a) uses a contrastive sequence calibration loss to steer the LM towards desired output. Rank responses to align human feedback (RRHF, Yuan et al., 2023) adds a ranking loss to the SFT loss. The ranking loss promotes responses based on preference ranked by humans or a reward model. Direct Preference Optimization (DPO, Rafailov et al., 2023) optimizes language models by contrasting it against a reference model on preference data. Rafailov et al. (2023) also provide a theoretical analysis that the DPO is optimizing the same objective as RLHF, but in a more efficient and stable manner. In our paper, we conduct empirical studies to compare offline post-training methods, RLHF, SLiC and DPO, in terms of performance and efficiency.

Human preference is expensive to collect thus difficult to scale up. Recently, there have been attempts to automate post-training by replacing the human preference data with model-generated feedback. Self-distillation with feedback (SDF, Xu et al., 2023b) samples multiple outputs from the model and prompts ChatGPT to pick the best response for fine-tuning the model. RL from AI Feedback (RLAIF, Lee et al., 2023) uses an off-the-shelf LLM to replace human labels in the standard RLHF. Following that, reinforcement learning from contrast distillation (RLCD, Yang et al., 2023) constructs model-generated contrastive pairs by prompting an off-the-shelf LLM to act differently on certain properties, e.g., harmlessness and helpfulness. Different from these works, our approach is an offline algorithm, which does not require time-consuming sampling during training. Our approach does not require training a reward model and can be easily scaled up.

## 3 PRELIMINARIES

**Reinforcement Learning from Human Feedback (RLHF)** To optimize the human preference with reinforcement learning, we need to first train a reward model $r_\tau(y|x)$ that outputs a reward for a given output $y$. When training the target model, RLHF (Ziegler et al., 2019) uses a reinforcement learning algorithm (usually PPO, Schulman et al., 2017) to optimize the reward of a sampled output $y$ from the target model $P_\theta$. To regularize the optmization and prevent model degeneration, a KL penalty term between the sequences of distributions over tokens of the target model and a reference model (e.g., SFT model) is added to the reward (Korbak et al., 2022). This prevents the RL policy from deviating substantially away from the reference model, which often leads to incoherent text output (Ziegler et al., 2019).

**Sequence Likelihood Calibration (SLiC)** In contrast to RLHF, SLiC can exploit pairwise human feedback data and train offline (i.e., without sampling from the target model each time). SLiC takes a positive example $y^+$, a negative example $y^-$ and a reference output $y_{ref}$ from the SFT model. In essence, SLiC encourages the target LM to output sequences those resemble the positive sequence

and penalizes those that resemble the negative sequence, while using the reference sequence from the SFT model for regularization. The loss function for SLiC is:

$$\mathcal{L}_{\text{SLiC}}(\theta) = \max(0, \delta - \log P_\theta(y^+|x) + \log P_\theta(y^-|x)) - \lambda \log P_\theta(y_{ref}|x) \tag{1}$$

where $\delta$ and $\lambda$ are two hyperparameters, controlling the margin for the ranking loss and regularization weight. SLiC is memory-efficient, as both its positive-negative pairs and reference sequences are offline.

**Direct Preference Optimization (DPO)**   Similar to SLiC, DPO is an offline preference optimization method. DPO takes a pair of (pre-computed) positive and negative examples and optimizes the difference between the target model and the reference model (i.e., SFT model), which increases the likelihood of the positive example and decreases the likelihood of the negative example. The loss function of DPO is shown below:

$$r^+(\theta) = \beta(\log P_\theta(y^+|x) - \log P_{ref}(y^+|x)) \tag{2}$$

$$r^-(\theta) = \beta(\log P_\theta(y^-|x) - \log P_{ref}(y^-|x)) \tag{3}$$

$$\mathcal{L}_{\text{DPO}}(\theta) = -\log \text{sigmoid}(r^+(\theta) - r^-(\theta)) \tag{4}$$

where $\beta$ is a temperature hyperparameter; $r^+$ and $r^-$ are the two pseudo-rewards that resemble the reward function in RLHF. Despite DPO having a similar form, there are key differences between SLiC and DPO: at train time, SLiC requires only the sampled outputs from a reference model, while DPO requires the logits from that (frozen) reference model for both the positive and negative sequence. Rafailov et al. (2023) also conduct a theoretical analysis of DPO and prove that optimizing the DPO loss is identical to the RLHF loss.

## 4   Contrastive Post-training over Pairwise Data Curriculum

**Contrastive Post-training**   Contrastive post-training involves the construction of positive $y^+$ and negative $y^-$ sequences in response to the same input $x$. Under the traditional settings of human-feedback, it is often the case that for some $(y_1, y_2) \sim P(x)$ sampled from the *same* LLM, human annotators provide a preference as to which is the positive. As this process is expensive, to reduce costs, recent studies (Xu et al., 2023b; Lee et al., 2023; Yang et al., 2023) have investigated the use of pre-aligned models as substitutes for human annotators in providing feedback for post-training methods. However, annotating preference pairs using the largest models, such as GPT-4, on datasets with millions of examples — like the 5M examples used by Orca (Mukherjee et al., 2023) — would incur a cost of $150k just for calling the API, making it prohibitively expensive as well.

In our setting, we choose to sample $y^+$ directly from a "superior" LLM, $y^+ \sim P_{sup}$, and $y^-$ from an inferior $P_{inf}$. We define one model to be superior to another $P_{sup} \succ P_{inf}$ if in expectation humans would prefer $y^+$ over $y^-$ given a reasonable input $x$. Relying on results in tried-and-tested benchmarks (Zheng et al., 2023; Li et al., 2023; Xu et al., 2023a) such as Alpaca Eval (shown in Table 1), we make an informed choice that GPT4 $\succ$ ChatGPT $\succ$ InstructGPT for our chosen scenario of general instruction tuning.

We acknowledge that there could be many reasons why humans would prefer $y^+$, as previous studies have found that a single reward function may not be sufficient to capture the range of human preferences (Hong et al., 2023; Skalse et al., 2023). Other studies emphasize only a certain property in the contrastive pair, such as helpfulness or harmlessness (Bai et al., 2022a).

**Data Curriculum**   The concept of a curriculum (Bengio et al., 2009) is analogous to the pedagogical approach in human learning where tasks are presented in increasing order of difficulty. By adopting this methodology, we aim to facilitate a smoother and more effective learning trajectory for our models.

For our curriculum, we approximate the difficulty of the learning task as being inversely proportional to the gap between the $P_{sup}$ and $P_{inf}$, as indicated in Table 1. That is, the more clear-cut the preference between juxtaposed $y^+$ and $y^-$, the easier the learning task. We define an `EasyPair` as $y^+ \sim$ GPT-4$(x)$ and $y^- \sim$ InstructGPT$(x)$. On the other hand, a `HardPair` contrasts between

Table 2: Time for post-training LLaMA-7B on Alpaca for one epoch on 16 Nvidia V100 GPUs.

| Method | SFT | RLHF/RLAIF (RM) | RLHF/RLAIF (PPO) | SLiC | DPO |
|---|---|---|---|---|---|
| Training Time | 4h | 3h | 24h | 7h | 12h |

e.g., ChatGPT and InstructGPT because the capability gap between them is narrower than that between GPT-4 and InstructGPT. `HardPairs` present a more nuanced challenge, requiring the model to discern subtler distinctions in quality and content.

We define our curriculum such that, initially, training starts with only `EasyPairs` to provides our model with a foundational understanding of the contrastive differences. During training, the model becomes adept at identifying distributional differences, so the probability of seeing an `EasyPair` in a mini-batch decreases as they are replaced by `HardPair`.

$$p(\texttt{EasyPair}) = 1 - \alpha$$
$$p(\texttt{HardPair}) = \alpha \tag{5}$$

As training progresses, $\alpha$ varies according to $f(t)$. In our experiments, we allow $f(t) = kt$ to be a linear function of the step number, or in some cases a constant function, for comparison. For the linear function, we choose $k$ such that $f(t) = 1$ at the end of one epoch, as shown in Figure 2. The anti-curriculum is the exact opposite – moving from `HardPair` to `EasyPair`.

We also explore an analogous curriculum regime for supervised fine-tuning, which we define as starting from ChatGPT targets (which are easier for a smaller model to imitate), and gradually moving towards GPT-4 targets, which are more challenging. By structuring such data curriculums, we ensure that the model can gradually acclimatize to the task, building on its understanding and refining its discernment capabilities. This approach not only enhances the model's performance but also provides insights into the incremental learning capabilities of large language models.

## 5 EXPERIMENTS

### 5.1 EXPERIMENTAL SETTINGS

**Training Datasets** Our small-scale experiments utilize Alpaca (Taori et al., 2023), an instruction learning dataset, which originally includes 52k instructions generated with Self-Instruct (Wang et al., 2023), with responses from InstructGPT (`text-davinci-003`). We further collect ChatGPT's responses with OpenAI API (`gpt-3.5-turbo`) and GPT-4's responses from Peng et al. (2023). Therefore, we are able to construct three contrastive pairs, namely *GPT-4 vs. td003*, *GPT-4 vs. ChatGPT* and *ChatGPT vs. td003*. For large-scale experiments, we use a mixture of 550k FLAN-v2 data, 200k FLAN-v1 data (sampled according to (Mukherjee et al., 2023)), the 52k Alpaca data (Taori et al., 2023) and 50k Vicuna data (Chiang et al., 2023).

**Evaluation Datasets** We evaluate performance of models with Alpaca Eval (Li et al., 2023) and the test set of WizardLM prompts (Xu et al., 2023a). Alpaca Eval consists of 805 instructions, which includes 252 instructions from the self-instruct evaluation set (Wang et al., 2023), 188 from Open Assistant evaluation set, 129 from Anthropic-HH helpfulness (Bai et al., 2022a), 80 from Vicuna evaluation (Chiang et al., 2023), and 156 from Koala evaluation (Geng et al., 2023). The metric is a win rate of a treatment candidate against a baseline model's responses, evaluated by GPT-4 in a side-by-side fashion (OpenAI, 2023).

The WizardLM test set (Xu et al., 2023a) consists of 218 prompts which cover 29 distinct skills, collected from the open-source repositories, platforms and forums. Following Xu et al. (2023a), we report the ratio of the sum over all examples of scores of the treatment model compared to a baseline (a.k.a. "score %") as well as the win/tie rates. This metric is again a side-by-side comparison evaluated by GPT-4. Whereas AlpacaEval formats comparisons as a ranking task (re-order the candidate responses according to how a human would prefer them), for WizardLM the candidates are individually scored. Note that such evaluation by GPT-4 might slightly favor SFT on GPT-4

Table 3: An example of reward hacking in RLAIF model trained with a "in-domain" reward model on GPT-4 vs. td003 pairs (Skalse et al., 2022), despite its response is unreadable.

| Prompt | Method | Response | Reward |
|---|---|---|---|
| Transform this recipe for "vegetable fried rice" into a vegan version. 3 tablespoons vegetable oil, 2 eggs, 1 cup diced onions, 2 garlic cloves minced, 2 cups shredded carrots, 2 cups cooked white rice, 2 tablespoons soy sauce. | RLAIF | 1/: BBCRed pepper converted to3 tbps shred blocklijke diceda)"Rotisserie vegetables" | 34.594 |
| | SFT | Here's a vegan version of vegetable fried rice: Ingredients: 3 tablespoons vegetable oil; 1 cup diced onions.. *[complete output omitted]* | 22.156 |

outputs, as pointed by Li et al. (2023). Both datasets have a different data distribution from our training set and thus can be a good testbed to test the zero-shot generalization capability of the models.

**Base Models**    For experiments on Alpaca, we use LLaMA-7B (Touvron et al., 2023a) as the base model. For large-scale experiments, we explore the post-training enhancement setting, where we initialize from 13B parameter state-of-the-art instruction-following model, Orca (Mukherjee et al., 2023) and improve its performance.

**Training Details**    For all model trained, we use the AdamW optimizer with a learning rate of 1e-5 and linear warm-up. The LLaMA models are trained on 16 Nvidia V100 32GB GPUs with the maximum length set to 1024 and a total batch size of 512. The Orca models are trained on 32 Nvidia A100 80GB GPUs with the maximum length set to 2048 and a total batch size of 512. The small scale experiments thus have 101 steps per epoch on Alpaca, and the large scale experiments have roughly 1600 steps. To save VRAM, we use DeepSpeed ZeRO-3 (Rajbhandari et al., 2020) for model parallelism and offload. For SLiC, we set the ranking margin $\delta$ and regularization coefficient both to 1.0, following Zhao et al. (2023a). For DPO, we use the default temperature $\beta$ of 0.1, following Rafailov et al. (2023). The training time for all methods on Alpaca is shown in Table 2. We implement RLAIF (Lee et al., 2023) by training reward models (initialized from LLaMA) with the same pairs for SLiC and DPO. Then, we use the trained reward models for the standard RLHF, strictly following Hugging Face TRL[2]. We search the KL penalty coefficient hyperparameter over $\{0.2, 0.5, 1.0\}$.

## 5.2    COMPARING CANDIDATES FOR POST-TRAINING: RLAIF, SLiC AND DPO

We compare offline contrastive post-training algorithms, SLiC and DPO, and an online RL method, RLAIF, to SFT. Since both Alpaca Eval and WizardLM evaluations are pairwise, we choose two reasonable baselines to compare all techniques: SFT on ChatGPT outputs, and SFT on GPT-4 outputs, which is slightly harder.

**Which is the best for post-training?**    The top of Table 4 establishes our baselines: we fine-tune LLaMA (Touvron et al., 2023a) on both ChatGPT and GPT-4 outputs, respectively. SFT on GPT-4 outperforms SFT on ChatGPT with a win rate of 61.2% and 72.7% on Alpaca and WizardLM evaluation sets, respectively.

For contrastive post-training approaches, SLiC underperforms SFT by a large margin. A potential reason is the objective that SLiC optimizes includes a fixed ranking margin $\delta$. In our setting, the distance between the positive and negative examples fluctuates, thus may cause difficulties for learning effectively. In contrast, DPO introduces a reference model instead of using a fixed margin for the loss. By comparing Equation 1 to Equation 4, DPO can be roughly regarded as optimizing a dynamic margin $\delta' = \log P_{ref}(y^+|x) - \log P_{ref}(y^-|x)$ as in SLiC. This may explain why DPO is more robust in our setting where the labels are noisy. Moreover, as shown in Table 2, DPO holds an advantage against RLAIF in training efficiency and alleviates the need to tune the hyperparameter

---

[2]https://github.com/huggingface/trl

Table 4: Experimental results of offline post-training techniques. For SLiC and DPO, the training target contrasts a positive vs. negative pair, and the reference model for these techniques is the SFT model trained on ChatGPT responses. All baselines are compared against LLaMA models fine-tuned with ChatGPT and GPT-4 responses on Alpaca data. SFT-3.5 is the LLaMA model trained with SFT on ChatGPT responses. [†]RLAIF-trained models suffer crippling reward hacking.

| Method | Init. | Training Target | Epoch | vs. SFT on ChatGPT | | | vs. SFT on GPT-4 | | |
| | | | | Alpaca | WizardLM | | Alpaca | WizardLM | |
| | | | | win% | score% | win (tie)% | win% | score% | win (tie)% |
| SFT | LLaMA | ChatGPT outputs | 1 | 50.0 | 100.0 | 50.0 | 37.4 | 97.4 | 32.4 (6.5) |
| SFT | LLaMA | GPT-4 outputs | 1 | 61.2 | 125.8 | 72.7 (6.0) | 50.0 | 100.0 | 50.0 |
| SFT | SFT-3.5 | GPT-4 outputs | 1 | 65.1 | 124.3 | 71.3 (5.1) | 53.2 | 103.8 | 47.2 (6.5) |
| RLAIF[†] | LLaMA | RM on output pairs | 1 | 0.0 | - | 0.0 (0.0) | 0.0 | - | 0.0 (0.0) |
| SLiC | LLaMA | ChatGPT vs td003 | 1 | 33.7 | 95.8 | 40.9 (0.5) | 20.5 | 85.9 | 24.5 (0.5) |
| SLiC | LLaMA | GPT4 vs ChatGPT | 1 | 41.3 | 108.8 | 57.9 (0.5) | 30.4 | 95.1 | 38.0 (0.9) |
| SLiC | LLaMA | GPT4 vs td003 | 1 | 22.9 | 81.4 | 31.0 (1.4) | 13.8 | 75.3 | 17.6 (1.4) |
| DPO | LLaMA | ChatGPT vs td003 | 1 | 48.6 | 111.3 | 58.8 (0.5) | 32.8 | 97.8 | 39.4 (0.5) |
| DPO | LLaMA | GPT4 vs ChatGPT | 1 | 56.0 | 119.6 | 68.1 (0.5) | 41.6 | 98.3 | 39.8 (1.9) |
| DPO | LLaMA | GPT4 vs td003 | 1 | 59.6 | 121.1 | 68.1 (2.8) | 45.2 | 99.8 | 43.1 (3.7) |
| DPO | SFT-3.5 | GPT4 vs td003 | 1 | 70.4 | 120.4 | 66.2 (2.8) | 58.7 | 105.4 | 51.9 (2.8) |
| SFT | SFT-3.5 | GPT4 outputs | 3 | 72.8 | 119.3 | 64.4 (4.6) | 62.1 | 103.4 | 48.1 (4.6) |
| DPO | SFT-3.5 | GPT4 vs td003 | 3 | **77.3** | **137.8** | **80.6** (1.9) | **66.5** | **112.2** | **62.5** (2.3) |

Table 5: Experimental results of RLHF compared with SFT and DPO. SFT-3.5 is the LLaMA model trained with SFT on ChatGPT responses.

| Method | Init. | Training Target | vs. SFT on ChatGPT | | | vs. SFT on GPT-4 | | |
| | | | Alpaca | WizardLM | | Alpaca | WizardLM | |
| | | | win% | score% | win (tie)% | win% | score% | win (tie)% |
| SFT | SFT-3.5 | GPT-4 outputs | 65.1 | 124.3 | **71.3** (5.1) | 53.2 | 103.8 | 47.2 (6.5) |
| DPO | SFT-3.5 | GPT4 vs td003 | **70.4** | **120.4** | 66.2 (2.8) | **58.7** | **105.4** | **51.9** (2.8) |
| RLHF | SFT-3.5 | OASST DeBERTa RM | 36.1 | 91.0 | 26.9 (7.9) | 25.3 | 86.6 | 22.2 (3.7) |
| RLHF | SFT-3.5 | OASST Pythia RM | 36.1 | 92.7 | 30.6 (9.7) | 29.4 | 87.9 | 25.5 (2.8) |

$\delta$. When comparing head-to-head with SFT on GPT-4 responses, the best-performing DPO wins on 58.7% and 51.9% prompts on Alpaca Eval and WizardLM, respectively.

**Which pair should we train DPO on?** We train multiple DPO models on different contrastive pairs. We find that the most distant pair, i.e., GPT-4 vs. InstructGPT, has the best performance. This may be due to this pair has the least noise, as most GPT-4 responses are expected to outperform those of InstructGPT. This provides a more reliable signal to facilitate model learning. As shown in Table 4, the DPO model trained on GPT-4 vs. InstructGPT outperforms the other two pairs on both Alpaca Eval and WizardLM evaluation. Also, we find that the DPO model initialized from the SFT model can achieve better performance than initialized from the raw LLaMA checkpoint.

**What if we train the model for even longer?** Due to computation budget limit, our previous experiments train the model for 1 epoch on Alpaca. However, we are curious if the advantage of DPO holds with more epochs training. We train both SFT and DPO models with 3 epochs, which is the same setting as in Alpaca (Taori et al., 2023) and Vicuna (Chiang et al., 2023). DPO keeps its advantage against SFT after 3 epochs and the performance gain is even larger. This DPO model outperforms SFT on all evaluation metrics by a large margin. This result suggests that DPO may be suitable for scaling up, which we will demonstrate later in Section 5.4.

Table 6: Head-to-head comparison of Orca 13B models in scaled-up experiments. Orca with DPO post-training significantly outperforms continuing training Orca with SFT ($p < 0.01$).

| Model | vs. | Alpaca Eval (win%) | | | | | | WizardLM Eval | |
|---|---|---|---|---|---|---|---|---|---|
| | | helpful | koala | oasst | self-instruct | vicuna | overall | score% | win (tie)% |
| Orca 13B | ChatGPT | 55.8 | 53.2 | 47.9 | 41.7 | 73.8 | 50.8 | 94.7 | 42.1 (16.9) |
| Orca + SFT | ChatGPT | 46.5 | 55.8 | 48.9 | 41.7 | **77.5** | 50.4 | 97.2 | 51.0 (11.9) |
| Orca + DPO | ChatGPT | **58.1** | **57.7** | **52.7** | **47.6** | 73.8 | **55.0** | **97.4** | 51.0 (11.1) |
| Orca + SFT | Orca 13B | 43.4 | **51.3** | 51.1 | 52.4 | 47.5 | 49.9 | **105.6** | **55.9** (19.9) |
| Orca + DPO | Orca + SFT | **59.7** | 48.7 | **60.6** | **56.0** | **51.3** | **55.8** | 104.8 | **55.9** (19.9) |

## 5.3 COMPARISON WITH RLAIF AND RLHF

For RL, we utilize three reward models: two external RLHF reward models from OpenAssistant reported in Table 5, and one RLAIF reward model trained "in-domain" on the contrastive pairs in the Alpaca dataset in Table 4. We strictly follow the settings and code implementation in Hugging Face TRL[3] library and use PPO to tune the SFT model on ChatGPT with 1 epoch with three different KL penalties coefficient {0.2, 0.5, 1.0} and report the best result among the three.

We find that PPO is unfortunately very sensitive to the quality of its reward model, and is prone to degeneration when trained on small amounts of possibly noisy "in-domain" data. An example is shown in Table 3, where a broken response trained with PPO is preferred over a coherent response generated by the SFT model. We believe this "reward hacking" is due to the reward model failing to generalize (Tien et al., 2023), likely overfitting to spurious lexical differences between GPT-4 and InstructGPT (Zhuang & Hadfield-Menell, 2020; Skalse et al., 2022).

To combat this behavior, we employ external reward models from Open Assistant (Köpf et al., 2023) which stabilize the training in the same codebase with the same settings off-the-shelf. In particular, we use the OpenAssistant DeBERTa-Large reward model[4] and the larger Pythia 6.9B reward model[5]. As Table 5 shows, while the outputs are coherent under these external reward models, they still fail to beat the SFT baselines, as the performance degrades on the two out-of-distribution evaluation datasets. This suggests the reward models may fail to generalize to out-of-distribution data (Tien et al., 2023). We conclude only that RLAIF/RLHF requires substantial effort to train properly. It is worth mentioning that DPO, as an alternative, works out-of-the-box on the *same pairs* that are used to train the "in-domain" reward models that lead to RLAIF's collapse.

## 5.4 ORCA+: SCALING UP CONTRASTIVE POST-TRAINING

To verify if our findings on small-scale Alpaca experiments can generalize, we test the performance of DPO with Orca 13B (Mukherjee et al., 2023) as both the reference model and initialization. The results are shown in Table 6. The SFT baseline is Orca trained on GPT-4 responses for the same prompts. The DPO model is trained with GPT4-vs-td003 pairs. We compare Orca 13B, Orca+SFT and Orca+DPO against ChatGPT responses. Orca+DPO can successfully improve the performance, achieving 55% win rate on Alpaca Eval and 51% win rate on WizardLM Eval, respectively. We then conduct a head-to-head comparison for SFT and DPO. Compared to the original Orca model, Orca+SFT does not show statistically significant improvement on Alpaca Eval ($p > 0.05$). Compared with Orca+SFT, Orca+DPO significantly improves performance on both Alpaca Eval and WizardLM Eval ($p < 0.01$). We also present generated examples in Appendix A. The large-scale experiments further verify the effectiveness of our proposed contrastive post-training approach.

---

[3]https://github.com/huggingface/trl

[4]https://huggingface.co/OpenAssistant/reward-model-deberta-v3-large-v2

[5]https://huggingface.co/OpenAssistant/oasst-rm-2-pythia-6.9b-epoch-1

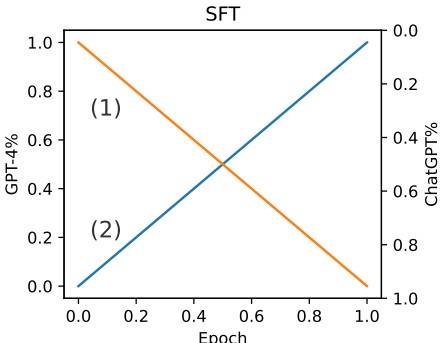 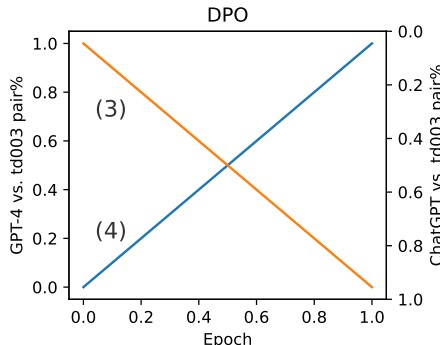

Figure 2: The four candidate data curriculums for SFT and DPO. For SFT *(left)*, the curriculum (1) fine-tunes the model on GPT-4 responses and gradually transitions to ChatGPT and the other (2) does the opposite. For DPO *(right)*, the curriculum (3) starts with GPT-4 vs. td003 and ends with ChatGPT vs. td003 while the curriculum (4) does the opposite.

Table 7: Experimental results of different curriculums for SFT and DPO. The corresponding curriculums are illustrated in Figure 2. SFT-3.5 is the LLaMA model trained with SFT on ChatGPT responses. Starting with `EasyPair` and warming up to `HardPairs` can significantly improve the performance compared to the best DPO model trained only with `EasyPair` (GPT-4 vs. td003).

| Curr. | Method | Init. | Training Target | vs. SFT on ChatGPT | | | vs. SFT on GPT-4 | | |
|-------|--------|-------|-----------------|--------|--------|------------|--------|--------|------------|
| | | | | Alpaca | WizardLM | | Alpaca | WizardLM | |
| | | | | win% | score% | win (tie)% | win% | score% | win (tie)% |
| (1) | SFT | LLaMA | GPT-4→ChatGPT | 47.5 | 107.6 | 52.8 (7.9) | 33.2 | 96.0 | 34.7 (2.3) |
| (2) | SFT | LLaMA | ChatGPT→GPT-4 | 57.0 | 115.2 | 59.7 (6.0) | 43.7 | 100.0 | 41.7 (4.2) |
| | SFT | SFT-3.5 | GPT-4 outputs | 65.1 | 124.3 | 71.3 (5.1) | 53.2 | 103.8 | 47.2 (6.5) |
| | DPO | SFT-3.5 | GPT4 vs td003 | 70.4 | 120.4 | 66.2 (2.8) | 58.7 | 105.4 | 51.9 (2.8) |
| (3) | DPO | SFT-3.5 | (GPT4→ChatGPT) vs td003 | **72.5** | 126.7 | 71.3 (2.3) | **59.8** | **108.9** | **57.4** (2.3) |
| (4) | DPO | SFT-3.5 | (ChatGPT→GPT4) vs td003 | 68.8 | **127.0** | **74.1** (3.2) | 56.8 | 105.2 | 47.4 (4.2) |

## 5.5 Data Curriculums for Post-training

We number different curriculums as shown in Figure 2. The experimental results for curriculums are shown in Table 7. All experiments are trained with the same numbers of contrastive pairs and steps. For SFT, starting with ChatGPT and transitioning to GPT-4 (Curr. 2) outperforms the opposite (Curr. 1) by a considerable margin. Since many models, such as Vicuna (Chiang et al., 2023) and Orca (Mukherjee et al., 2023), are fine-tuned with mixed ChatGPT and GPT-4 responses, our finding suggests that a simple reordering of the data can lead to different performance.

For DPO, with Curr. 3, we start from `EasyPair`, GPT-4 vs. td003 and transition to `HardPair` Chat-GPT vs. td003. This strategy achieves better performance than using only `EasyPair` all the time. Meanwhile, the anti-curriculum, Curr. 4, underperforms single-pair DPO in general. Curriculum learning further unleashes the potential of DPO for post-training. We believe further improvement can be achieved with more thorough hyperparameter search.

## 6 Conclusion and Future Work

In this paper, we propose a new setting for contrastive post-training large language models. We explore the best method and curriculum settings to facilitate post-training. Our large-scale experiments with a state-of-the-art model Orca further verify the effectiveness of our approach and suggest its potential for improving performance of LLMs at scale. For future work, we plan to explore both how to better select meaningful contrastive pairs from fixed data regime, and subsequently to continually learning evolving a model with pairs populated by sampling from the model itself at various points through training.

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

## A EXAMPLES OF GENERATED RESPONSES

| Instruction #1 | How did US states get their names? |
|---|---|
| LLaMA SFT on GPT-4 | The official name of a state comes from the name of the state's capital city, followed by an additional descriptive suffix. For example, Texas is named after the city of San Antonio, while California is named after the city of San Diego, and New York is named after the city of New York. In some cases, the state name is also derived from the name of a prominent natural feature or the name of a city that no longer exists. For more information about the history of US state names, visit the website of the US Census Bureau at www.census.gov. |
| LLaMA Curriculum DPO (Curriculum 3) | There are many stories and myths behind the naming of US states. Here are some popular ones:
1. Alabama: Named after the Native American tribe known as "Alabama."
2. Alaska: Named after the Aleutian word "Alyeska" which means "great land."
3. Arizona: Named after the Spanish word for "arizona," which means "little spring."
4. Arkansas: Named after the Sioux word "as-kah-ras-sha," which means "land of downriver people."... |
| Orca | US states have diverse origins for their names, which include indigenous words, European languages, geographical features, historical events, and personal names. Here's a brief overview of how some states got their names:
1. Alabama: Derived from the Native American tribe called the Alabama, meaning "thicket breakers" or "cotton pickers."
2. Alaska: Based on the Aleut word "Alaksa," which means "great land."
3. Arizona: Derived from the Native American word "Arivipa," meaning "small spring."
4. Arkansas: Originally "Arcahaise," meaning "land of the south wind" in the Native American Quapaw language... |
| Orca with DPO | The naming of U.S. states is a diverse and fascinating history that reflects the cultural, historical, and geographical influences of the various groups of people who settled and governed the land. Here's a brief overview of how some U.S. states got their names:
1. Virginia: Named after the Virgin Queen, Queen Elizabeth I of England, who was known as the "Virgin Queen" because she never married and remained a virgin. The state was founded in 1607 by the Virginia Company, a joint-stock company chartered by King James I.
2. Massachusetts: The name comes from the Massachusett tribe, one of the Native American tribes that inhabited the region. The name translates to "large hill" or "place of big rocks."
3. Pennsylvania: Named after the Pennsylvania Colony, a British colony that was established by William Penn, an English Quaker and founder of the City of Brotherly Love (Philadelphia). The name Pennsylvania combines the name of the colony's founder with the Latin word "Penn" (meaning "hill" or "top"), which refers to the colony's geographical location on the Pennines, a range of hills and mountains in the Appalachian Mountains.
4. Maryland: Named after the wife of King Charles I of England, Queen Henrietta Maria, who was known as the "Maria" in her name. The "land of Mary" is derived from her name, and the "d" at the end is an Old English suffix that indicates "territory of." ... |

| Instruction #2 | How do I wrap a present neatly? |
| --- | --- |
| LLaMA SFT on GPT-4 | To wrap a present neatly, you can start by choosing the right paper for the occasion. White, kraft or black paper are common choices for wrapping presents. Then, you can cut the paper to size, wrapping it around the present with the paper folded in half lengthwise. You can also use tape to secure the ends of the paper. For a more professional look, you can add a bow or a ribbon to the top of the package, and write a message or add a tag. |
| LLaMA Curriculum DPO (Curriculum 3) | To wrap a present neatly, follow these steps:
1. Select a good-quality paper or wrapping paper that matches the color scheme or theme of the present.
2. Lay the present on a flat, clean surface. If it's rectangular, place it on the left side, and if it's cylindrical, place it on its side.
3. Fold the wrapping paper in half, at the center crease, then in half again, making a triangular fold at the center crease.
4. Fold the right side of the triangle over the center crease, leaving about 1-inch of the right side exposed.
5. Fold the bottom of the triangle up, so that it touches the center crease, then tuck the right side into the left side of the package.
6. Take a small piece of tape and wrap it around the right side of the package, securing it to the left side of the package. Repeat on the other side of the package.
7. Wrap the present with the top of the package first, folding the wrapping paper over the top, and tucking it under the box.
8. Secure the end of the wrapping paper with more tape, and wrap the rest of the package, leaving a little space at the bottom to tuck in the ends.
9. Fold the ends of the wrapping paper over the box, and tape or secure them with a rubber band.
10. If you want to add a bow or ribbon, place it on top of the package, and secure it with tape or a rubber band.
By following these steps, you can wrap a present neatly and create a nice presentation for your gift. |
| Orca | Wrapping a present neatly involves a few simple steps:
1. Choose the right materials: Select a clean, large piece of wrapping paper or gift bag that is proportional to the size of the gift. You may also use tissue paper, fabric, or reusable bags for a more eco-friendly option.
2. Measure and cut the paper: Measure the dimensions of the gift (length, width, and height) and add extra paper or bag length to allow for folding and tucking. Cut the paper or fabric to size, leaving enough extra for folding and tape.
3. Place the gift in the center: Lay the gift flat in the center of the paper or fabric. If you're using a gift bag, place the gift on top of the bag and ensure it's centered.
4. Fold the sides: Fold the sides of the paper or fabric over the gift, aligning the edges with the sides of the gift. For a more polished look, fold the paper or fabric inward by about an inch and then fold it back out for a crisp edge.
5. Fold the bottom: Fold the bottom of the paper or fabric up over the gift, leaving enough excess for the bow or ribbon. Ensure the fold is even and the gift is snug within the folded paper or fabric.
6. Secure the wrap with tape: On the top and bottom edges of the wrap, apply double-sided tape or a small piece of clear packing tape for a clean hold. If you're using a gift bag, ensure the handles are folded over and secured with tape.
7. Add a decorative touch: Add a ribbon, bow, or gift tag to personalize the present. If using a gift bag, you can add a handle wrap or a fabric ribbon for a more polished look.
8. Finish with a gift holder: Use a non-slip gift holder or shipping weight to keep the wrapped gift stable and secure during transport.
Remember to keep the wrapping neat and tidy, with clean folds and even tape application. This will ensure your present looks beautiful. |

| | |
|---|---|
| Orca with DPO | Wrapping a present neatly can make the gift look more appealing and show the recipient that you put effort into your gift. Here's a step-by-step guide to wrapping a present neatly: |

1. Choose the wrapping paper: Select a wrapping paper that complements the gift's shape, size, and color. If the gift is rectangular, choose a paper that is wider than the width of the gift. If it's a cylindrical gift, choose a paper that is tall enough to wrap around it and overlap slightly.

2. Cut the paper: Use a pair of sharp scissors to cut the wrapping paper to the appropriate size. For a rectangular gift, cut the paper so that it's large enough to wrap around the gift and overlap by at least 2-3 inches (5-7.5 cm) on two opposite sides. For a cylindrical gift, cut the paper so that it's tall enough to wrap around the gift and overlap by at least 1-2 inches (2.5-5 cm) on two opposite sides.

3. Fold the paper: Lay the paper flat on a clean, flat surface. Fold the paper in half lengthwise (if it's a rectangular gift) or in half around its longest diameter (if it's a cylindrical gift). This will create a crease down the center of the paper, making it easier to fold and align the edges.

4. Position the gift: Place the gift in the center of the folded paper, aligning the edges with the crease. For a rectangular gift, the gift should be centered horizontally within the paper, and the edges of the paper should align with the top and bottom edges of the gift. For a cylindrical gift, the gift should be centered vertically within the paper, and the edges of the paper should align with the top and bottom edges of the gift.

5. Tuck and fold: Gently tuck the edges of the paper under the gift, folding them neatly around the sides and bottom of the gift. Make sure the paper is smooth and free of wrinkles. For a rectangular gift, tuck the top edge under the gift as well, and then fold the bottom edge up over the gift, creating a crisp edge. For a cylindrical gift, fold the paper around the top and bottom edges, creating a smooth, even edge.

6. Secure the paper: If the wrapping paper is slippery, you may need to use tape to secure the edges. Apply a small piece of tape on the inside edge where the paper meets the gift, and gently press the paper in place. Continue applying tape to other edges, making sure the paper stays smooth and wrinkle-free. For a more polished look, you can also use double-sided tape or a glue stick to attach the paper.

7. Add a decorative touch: To make your present look even more festive, consider adding a ribbon, gift tag, or a decorative element like a confetti pocket, bows, or decorative paper. Choose colors and patterns that complement the gift and the recipient's style.

8. Place the gift in a gift bag: If you want to keep the gift wrapped but still use a gift bag, simply place the gift inside the wrapped present. This can be especially helpful if the present is heavy or fragile.

9. Store the gift: Keep the wrapped present in a cool, dry place until it's time to give it to the recipient. Avoid storing it near direct heat sources or sunlight, which can cause the paper to wrinkle.

