# OpenReview forum: "Contrastive Post-training Large Language Models on Data Curriculum"
_ICLR.cc/2024/Conference — ICLR 2024 Conference Withdrawn Submission_

### Official Review · Reviewer_Vr47 · 2023-10-29

**Soundness:** 3 good
**Presentation:** 3 good
**Contribution:** 2 fair
**Rating:** 5
**Confidence:** 4

**Summary:**

This paper investigates contrastive post-training techniques designed to align large language models (LLMs) with human preferences, using contrastive preference pairs generated by text-davinci-003, ChatGPT and GPT-4. The central contribution is the evaluation of some prominent post-training methods such as SLiC, DPO, and SFT. This leads to some valuable conclusions for the post-training of LLMs. Furthermore, the paper proposes integrating a data curriculum learning scheme into DPO and SFT. Comprehensive experiments are conducted, revealing that a proper choice of the setting for contrastive post-training for LLMs is very beneficial.

**Strengths:**

This paper offers insightful recommendations for post-training LLMs using contrastive preference pairs generated by OpenAI's models. This method presents a cost-effective approach to aligning LLMs with human preferences without the need for expensive human annotations.

This paper introduces an interesting approach that integrates post-training techniques with a data curriculum learning scheme.

**Weaknesses:**

I’d like to see this paper as an empirical study of contrastive post-training for LLMs, offering extensive comparisons among different methods and demonstrating effective combinations of them. Given this, however, I'm not pretty sure the contribution possesses sufficient novelty for acceptance at ICLR. As it stands, venues focused on empirical methods might be more appropriate.

The paper's narrative appears disjointed. While the introduction emphasizes an affordable method of generating human preference data, the experimental section seems to underscore the comparison of post-training techniques. The title predominantly focuses on curriculum learning. Despite showing positive effects, the inclusion of curriculum learning seems unnecessary. The rationale behind combining curriculum learning with post-training is not extensively discussed.

The use of curriculum learning is not strongly related to contrastive post-training. While it is totally fine to introduce any methods into this work, This makes the focus of the work somewhat scattered.

The central theme of the paper is aligning LLMs with human preferences; however, evaluations solely rely on GPT-4, neglecting human evaluations.

**Questions:**

1. Table 4 suggests that the number of training epochs significantly impacts model performance. It appears that the model parameters haven't fully converged after just one epoch of training. Figure 2 shows that the curriculum-based experiments also operate on a one-epoch basis. Given that curriculum learning can potentially hasten model convergence, one wonders if the advantages of data curriculums arise from its training efficiency. If both models, with and without data curriculums, are trained over three epochs, would the performance of post-training with data curriculums still outstand?

2. The paper's direction seems ambiguous. If this work is to be perceived as innovative, the authors highlight in the introduction that using existing LLMs to generate offline preference data is both cost-effective and efficient. However, from this standpoint, the authors do not delve deeper in subsequent sections. For instance, the authors could have compared the effectiveness of preference data annotated by human with that generated by models, or they could have discussed potential noise issues in model generated preference data. Conversely, if this work is more analytical, the paper invests significant time in determining the superior post-training strategy, an emphasis missing from the introduction.

---

### Official Review · Reviewer_kAVD · 2023-10-31

**Soundness:** 2 fair
**Presentation:** 2 fair
**Contribution:** 1 poor
**Rating:** 3
**Confidence:** 2

**Summary:**

This work analyzes different contrastive learning techniques, i.e. SLiC and DPO for further fine tuning language models and compare them with the performance given from regular supervised finetuning.

In addition, an analysis on curriculum learning, varying the difficulty of the training pairs as part of the contrastive training is analyzed

Many experiments are carried out. Core experiments include fine tuning two versions of LlaMa 7B (vanilla and fine tuned with ChatGPT responses, called SFT-3.5). Many configurations are tested, including supervised finetuning using ChatGPT responses, GPT-4 responses, using SLiC and DPO techniques, on both LlaMa and SFT-3.5. Conclusions are provided, showing that DPO is the most effective technique for fine tuning, using stronger examples as positive ones is better and training for longer provides better results.
An ablation analysis shows that, with the same pairs, RLHF and RLAIF do not reach similar performance to what DPO is capable of.
Further experiments are carried out showing that the technique can be scaled up to a bigger language model such Orca+.
Finally, an experiment using curriculum learning, varying the quality of the answers to more powerful LM provides a further but minor improvement.

**Strengths:**

The work makes a great effort to cover many of the existing methods of fine tune alignment of Large language models from RLHF, RRHF, DPO and SLIC.
There is an extensive experimentation that shows different configurations for post training and existing language models.

**Weaknesses:**

The approach is built upon existing methods and the contribution seems to be relatively modest.
Authors claim that they propose “a new automatic setting for contrastive post-training” but they use existing contrastive techniques as SLiC and DPO. The novelty seems to be using different language models as a source of  positive / negative examples, but similar techniques were already used on reward models such as RRHF (Yuan et al 2023).

The abstract, introduction and conclusion are not completely aligned. While the abstract claims that authors “explore contrastive post-training techniques for alignment by automatically constructing preference pairs from multiple models of varying strengths”, both the intro and the conclusions claim that the authors propose “a new setting for contrastive post-training large language models”. Either authors propose existing techniques or propose a new one.

While the paper is titled “... on Data Curriculum”, this aspect plays a minor role in the entirety of this work. Only table 7 and sections 5.5 cover the experiments on this aspect. Only a single curriculum technique is explored (using 2 levels of hardness and varying the proportion through training). Results on this configuration are modest and only prove that curriculum works.

**Questions:**

This work seems like an analysis paper rather than a paper presenting a new method. Why did the authors not present the work in such a way?

What were the expectations from the authors? DPO is known to be more effective and stable than RLHF. Also SLiC.

---

### Official Review · Reviewer_7AN3 · 2023-11-01

**Soundness:** 3 good
**Presentation:** 3 good
**Contribution:** 3 good
**Rating:** 5
**Confidence:** 3

**Summary:**

This study proposed a contrastive post-training framework where leverages the various off-shelf LLMs to generate the synthetic contrastive pair data. Such synthetic data can be used in two types of existing contrastive technique: SLiC and DPO to serve as calibrator to further improve the performance of SFT.  Under large scale experiment settings, proposed method helps Orca to outperformed ChatGPT.

**Strengths:**

1. Most of contributions claimed are well-supported by extensive experiments results. The contrastive learning framework performed better then RLHF loss and SFT loss with the help of high quality of contrastive pairs from off-the-shelf LLMs.
2. The topic is very timely under the bigger conceptual umbrella: the model distillation from various LLMs sources.
3. This paper is well-written and easy to follow.

**Weaknesses:**

1. The novelty is limited. From framework perspective, the credit of contrastive loss has been taken by previous work (SLiC and DPO). From training data perspective, no new corpus is contributed, the credit has been taken by previous work LLM-generated contrastive corpus (Peng et al 2023.)
2. The contribution is overclaimed due to unfair comparison. Please refer the question to author.

**Questions:**

Though there is no RLHF reward model used, the LLMs (ChatGPT, GPT-4) has been trained with human alignment and RLHF. In other words, the generation process of contrastive pairs is inherited with RLHF.  Can you explain a little bit more about to support your contribution claims “improves performance of LLMs without human-, AI- or reward model-feedback”? Do you think this statement is over-claimed?

---

### Official Review · Reviewer_ohbY · 2023-11-06

**Soundness:** 2 fair
**Presentation:** 1 poor
**Contribution:** 2 fair
**Rating:** 3
**Confidence:** 2

**Summary:**

This paper studies the relationship between model-generated positive-negative pairs and contrastive post-training techniques. The authors generate positive-negative contrastive datasets by pairing outputs of pretrained LLMs of differing strengths. They run experiments on 7B LLAMA models to investigate the efficacy of using these datasets with post-training constrastive learning techniques to improve model performance. They scale up some of these experiments to Orca 13B. The authors present an analysis of ordering the data on an easy/hard curriculum.

**Strengths:**

The issue of human-in-the-loop post-training being expensive and unstable is a significant one for LLM training at large, so the subject of this work is well motivated. Many experiments are run, which provide lots of information comparing the methods and data setups.

**Weaknesses:**

The novelty of the work is unclear. It would help to state clearly which parts of the described methods/experiments are novel contributions.

Sec 5.2: The experiments presented to compare the methods do not run to convergence, so using them to compare the methods may not be entirely fair. As the primary contribution of the work is to run experiments comparing extant methods, issues in the comparison of those methods are especially impactful to the overall value of the work.

The paper has numerous significant issues in readability that can be easily resolved to improve the clarity of the work. Model names are used to refer to models and datasets produced by those models interchangeably and without warning. The experimental setup is complicated, and this serves to obfuscate it further, and makes reading the paper much harder than it needs to be. This is thankfully very easy to fix. The

Sec 5.5: The curriculum experiments are missing the baseline of a uniform distribution of the mixture of easy/hard datasets (ie training on the mixed datasets without any curriculum). Without this baseline it is impossible to assess the relative contribution of the curriculum itself, as opposed to just having a larger and more diverse dataset. Additionally, the curriculum experiments are missing analysis or discussion that might explain the value or implications of the results.

**Questions:**

Is the construction of positive/negative pair sequences via sampling from "superior" and "inferior" models a novel contribution? If so, please state as much. If not, please provide a citation.

Table 4 / Section 5.2."Which is the best for post-training?" : while it is understandable to have resource constraints on running large experiments with many models, the significant gap between the 1-epoch and 3 epoch SFT / DPO runs (65.1 -> 72.8 Alpaca win % for SFT-3.5 -> SFT on D_{GPT4}, and 70.4 -> 77.3 Alpaca win % for SFT-3.5 -> SFT on D^{GPT4}_{td003}) suggests that the model performance is far from convergence at 1 epoch. This makes it difficult to take seriously the relative performance of all the 1-epoch training runs presented in the table. This is especially true for the comparison of the different training methods, as it is highly plausible that their training curves may have different characteristics, making a 1-epoch comparison not necessarily indicative of final converged performance.

In Section 5.1 and onwards, model names are overloaded with *datasets* generated from those models. For example "GPT-4" is used to refer to the model previous to sec 5.1, but then afterwards is used to refer either to the model itself or the datasets generated from the model (ie. "GPT-4 vs. td003, GPT-4 vs. ChatGPT"). Consider doing something like *D*_{GPT-4} to refer to the dataset of gpt-4 generated examples, and *D*^{gpt4}_{chatgpt} to refer to the contrastive dataset with positive *D*_{GPT-4} and negative *D*_{chatgpt} outputs.

The use of X 'vs' Y is also overloaded, as occasionally it refers to head-to-head model output comparison (as in Table 1) and sometimes it refers to a contrastive dataset combining outputs of two models. This would be fixed by adopting the *D*^{X}_{Y} notation convention.

Additionally, in Sec 5.1 you introduce "td003" as an alternative name for InstructGPT without any comment as to the convention you are establishing there. Please say explicitly "We use td003 to refer to the dataset generated by InstructGPT" (or preferably *D*_{td003}) or something along those lines. Later on in Sec 5.2 (Which pair should we train DPO on?) you go back to using "InstructGPT" to refer to the dataset... "GPT-4 vs. InstructGPT".

Figure 1 cartoon could be modified to improve clarity. If you didn't know what RLHF or constarstive post-training were, it would be difficult to assess from the figure.

Figure 2 and Table 7 do not use the same nomenclature, which makes it difficult to parse what is going on. Table 7 refers to EasyPair and HardPairs, but Figure 2 does not feature either. It is difficult to parse what is going on in Table 7 due to the notational inconsistency. Please make much more clear which datasets (/curricula) are being used. Preferably by using "EasyPair" and "HardPair" rather than their constituent datasets, as the most relevant piece of information in the curriculum context is the relative difficult rather than the absolute source. Again, using D_{X} notation will significantly help here.

Table 4:
"SFT-3.5" sounds like a modification on SFT, which is a training method, rather than the model that is being modified by SFT. Maybe rename to LLAMA+ / LLAMA_{SFT} / LLAMA_{ChatGPT}? "3.5" being used as stand-in for ChatGPT without explanation or precedent is also confusing. Also, is the model referred to as SFT-3.5 the final trained model of the first row of the Table? If so, please clarify in the Table heading.

Also consider putting the 3-epoch runs below the 1-epoch runs that they correspond to, in order to improve the capacity to compare between the two. Is it possible the "crippling reward hacking" of the RLAIF setup can be alleviated with careful tuning?

Table 6: what is the bolding convention being used here? Please describe in the table header. Also, please include the comparison of Orca+DPO vs Orca 13B.

Nit:
Table 3: "despite its response is unreadable" --> "despite its response being unreadable"